# Characterisation of Extracts and Anti-Cancer Activities of *Fomitopsis pinicola*

**DOI:** 10.3390/nu12030609

**Published:** 2020-02-26

**Authors:** Karen S. Bishop

**Affiliations:** Discipline of Nutrition and Dietetics/Auckland Cancer Society Research Centre, School of Medical Sciences, Faculty of Medicine and Health Sciences, University of Auckland, 85 Park Road, Grafton, Auckland 1023, New Zealand; k.bishop@auckland.ac.nz; Tel.: +64-9-923-4471

**Keywords:** anti-cancer properties, extracts, *Fomitopsis pinicola*, location, medicinal history, sequence identification, taxonomy

## Abstract

*Fomitopsis pinicola* (Sw. Karst) is a common bracket fungus, with a woody texture. It is found predominantly in coniferous forests in temperate regions throughout Europe and Asia. *Fomitopsis pinicola* has been extensively used for medicinal purposes, particularly in Chinese and Korean traditional medicine. In this mini-review, the anti-cancer characteristics of *F. pinicola* extracts were investigated. In vitro experiments revealed the pro-apoptotic, anti-oxidant and anti-inflammatory properties of extracts, whilst two of three in vivo studies reported an inhibition of tumour growth and prolonged survival. Only studies wherein fungal specimens were sourced from Europe or Asia were included in this review, as samples sourced as *F. pinicola* from North America were probably not *F. pinicola*, but a different species. Although not one of the most revered fungal species, *F. pinicola* has been used as a medicinal fungus for centuries, as well as consumed as a health food supplement. To date, the results from only three in vivo studies, investigating anti-cancer properties, have been published. Further studies, using comprehensively identified specimens, are required to fully elucidate the anti-cancer properties of *F. pinicola* extracts.

## 1. Introduction

*Fomitopsis pinicola* (Sw. Karst), is a common woody fungus found in coniferous forests in temperate regions throughout Europe and Asia [1], including the Himalayas [2]. Numerous local names exist for *F. pinicola*, such as the Japanese name, which is Tsugasaruno-koshikake [3], and the English name of red-belted bracket fungus [4]. *Fomitopsis pinicola* is commonly known as a brown-rot fungus, characterised by bipolar sexual compatibility and the presence of the phenol oxidase, tyrosinase (with extracellular oxidase not present) [3]. It has been used in Chinese and Korean traditional folk medicine as an anti-inflammatory agent and for general well-being.

The fruiting body is fan shaped, has a hard, woody texture, can grow up to 40 cm in diameter (Figure 1), and is often referred to as the red belt conk. The fruiting body has a glossy appearance and can be red-brown or a lighter colour depending on the age of the specimen. It grows by adding an additional layer or tube annually. The fungus is saprobic and can also be parasitic, causing heart rot in living trees, and brown cuboidal rot in dead trees [2]. Decay fungi such as *F. pinicola* are often thought to be symbiotic and this could be due to the presence of fungi and nitrogen fixing bacteria at the same sites on fir trees [5]. In addition, they help circulate forest nutrients through the decay of dead tree trunks, although the brown rot residues can remain in the soil for extended periods before breaking down [6,7]. However, *F. pinicola* and other brown rot species can also contribute significantly to forestry loses, particularly at sites where the bark has been damaged as might occur when branches are removed.

*Fomitopsis pinicola*, like many other fungi, are predominantly identified phenotypically, but require molecular biology techniques to confirm the identification. Internal spacer region (ITS)2 sequencing is a suitable method that is routinely used for the correct identification of numerous species, including *F. pinicola*. Unfortunately, it can appear phenotypically similar to *Ganoderma lingzhi* and other species of the genus *Fomitopsis*, and therefore it is important to confirm the speciation of the specimen one is working with prior to publication.

A literature review of the anti-cancer properties of *F. pinicola* was performed using Embase, Web of Science and Google Scholar. Articles, published in English, where an in vitro and/or in vivo approach was implemented to investigate the anti-cancer properties of *F. pinicola* extracts, were included. Due to extensive fungal species misidentification [8], taxonomy and means of accurate identification of *F. pinicola* were also explored. Search terms included “*Fomitopsis pinicola*”+ “cancer” + “in vivo.” Thereafter “anti-inflammatory” was substituted for “cancer”, and an additional article was returned. In a similar manner “in vitro” was substituted for “in vivo”, and “*Fomitopsis pinicola*” + “taxonomy” were also searched. Pearly growing was implemented. This article is not a systematic review and, together with the implementation of pearly growing, it was decided not to include numbers and justification for article inclusion and exclusion.

## 2. The Taxonomy of *F. Pinicola*

Fungi are poorly, and sometimes incorrectly, described [8]. More recently, sequence-based classification and identification (SBCI) has been used to detect and classify environmental fungi and also to confirm or dispute identification or classification of named specimens. The ITS of rRNA genes can be PCR-amplified and sequenced, and this method is commonly used for SBCI [8]. Further, 16S rRNA sequences may also be used for this purpose, but it is regarded as less accurate than ITS sequencing, as the latter is less highly conserved and is therefore more likely to vary from one species to another [9]. To help avoid misidentifications, Edgar recommends the sequencing of two rather than one variable region, which could include V3, V4, V5 and ITS, or full-length 16s rRNA or large subunit rRNA genes [8,9]. With the integration and standardisation of stand-alone databases, and the incorporation of phylogenetic trees into pipelines used to identify or name specimens, data will be easier to incorporate into databases and therefore more likely to be deposited, and easier to access, thus strengthening the accuracy of fungal identification [8].

*F. pinicola*, an ancient polypore species, is classified according to the Integrated Taxonomic Information System [10] as follows:

Kingdom: Fungi

Division: Basidiomycota

Class: Agaricomycetes

Order: Polyporales

Family: Fomitopsidaceae

Genus: Fomitopsis

Species: *F. pinicola*

*Fomitopsis pinicola* was originally named in 1810 as *Boletus pinicola* by Swartz and then transferred to *Fomitopsis* by Petter Karsten in 1881 [11,12]. *Fomitopsis pinicola* (Swartz ex Fr.) P. Karst. (1881) was also named as *Polyporus pinicola* Fr. [3] before sequencing was used to clearly define the species. More recently, Binder et al. performed whole-genome sequencing using a shotgun approach, and classified *F. pinicola* in the antrodia clade [7].

In 2016, Haight et al. reported on an investigation into the suspected *F. pinicola* complex [14]. Based on samples collected in North America, Europe and Asia and phenotypically identified as *F. pinicola*, four distinct species were identified, with only *F. pinicola* found in Europe and Asia. The other three species were found in different regions of North America [1,14]. For this reason, articles based on samples collected outside of Europe and Asia were not included in this review article. 

*F. pinicola* is widely available and has been extensively used for medicinal purposes, particularly in Chinese traditional medicine [15]. However, the use of *F. pinicola* in Central European folk medicine has been largely forgotten [16]. Like many hardwood bracket fungi, it is believed that *F. pinicola* specimens were traditionally prepared for consumption as a soup/tea or in alcohol [15]. 

Although not one of the most revered fungal species, *F. pinicola* (Sw Karst) has been used as a medicinal fungus for centuries for the treatment of headaches, nausea and liver disease [16], as well as in health food supplements [15,16]. 

## 3. Active Ingredients

For centuries, medicinal mushrooms have been used by various cultures to enhance health. Pharmacologic research into medicinal mushrooms, using in vitro, in vivo and clinical studies, has been used to identify several health benefits and their associated biological pathways [17]. However, very little research has been carried out on *F. pinicola*. A variety of extraction methods, whole extracts, fractions and compounds isolated from the mushroom, have been tested. Many of these are listed in Table 1. Studies carried out on specimens sourced from North America were not include (e.g., Liu et al. [18])

Although Table 1 includes the compounds that were detected in *F. pinicola* using different extraction methods, the anti-cancer activities were not assessed. Various phytochemicals have been shown to have specific anti-cancer properties, but it is generally accepted that these compounds probably act synergistically to achieve an anti-cancer effect [25]. Wang et al. identified ergosterol in a chloroform extract from *F. pinicola* and observed anti-cancer properties such as a pro-apoptotic and inhibition of migration effects [26]. Further, in a study published by Yoshikawa et al., fomitopinic acids and fomitosides inhibited cyclooxygenase (COX) 1 and 2 activity [23]. Although many of the compounds detected in *F. pinicola* have not been assessed in isolation, some of the compounds have been isolated from other species and found to have anti-cancer properties e.g., gallic acid [27]. Based on the available evidence, it is not possible to determine exactly which compounds exert the strongest anti-cancer properties, and further research is required.

## 4. Anti-Cancer Activities

Medicinal properties of mushrooms, based on hearsay, have been recorded for thousands of years—for example, *Ganoderma lucidum* (Lingzhi) has been used for general well-being since before the 5^th^ century by the Chinese [28]; *Formes fomentarius* has been used as a potent anti-inflammatory agent by the Greeks (450 BC) [29]; and puffball mushrooms of the genus *Calvatia*, have been used for centuries by Native Americans to promote wound healing [29]. More recently, medicinal mushrooms have been used as an adjuvant to cancer therapy to enhance the effects of treatment and for the alleviation of side effects from chemo- and radiation therapy (e.g., nausea) [30]. Furthermore, numerous clinical trials have been conducted to assess the potential anti-cancer properties of both in-house and commercially prepared medicinal mushrooms [30]. Fewer than ten in vitro studies on cancer cell lines have been published, but the number of in vivo publications on *F. pinicola* are even more limited. 

### 4.1. In Vitro Studies

Numerous cell culture experiments have been used to investigate the anti-cancer properties of *F. pinicola* extracts. These studies have been outlined in Table 2.

Hanahan and Weinberg described various hallmarks of cancer [32], which have enabled us to study the impact of extracts/compounds on these hallmarks (e.g., evasion of programmed cell death) and their related pathways, rather than on cancer directly. Underlying these hallmarks are mechanisms such as inflammation, genome instability and the creation of a tumour microenvironment [32]. Many of the in vitro studies outlined in Table 2 showed an increase in anti-oxidant activity [31], increase in apoptosis [19,26] or an upregulation of pro-apoptotic genes [21,22], and anti-inflammatory activity [23]. In addition, PARP, which is involved in DNA repair, genomic stability and programmed cell death, increased in response to treatment in a sarcoma cell line [20]. Cell cycle dysregulation is another hallmark of cancer [32] and may be a target of the mechanism of action of FPKc. This reasoning is supported by in vitro evidence showing the inhibition of cell proliferation; damage to cell membrane in sarcoma but not healthy cells; the triggering of S-phase cell cycle arrest; a decrease in MMP and release of mitochondrial cytochrome C [19]. Together, these in vitro studies show that *F. pinicola* extracts/compounds have anti-cancer activities which warrant further investigation.

### 4.2. In Vivo Studies

A small number of in vivo studies have been performed wherein the anti-cancer properties of *F. pinicola* were investigated. These studies are listed in Table 3. In two of the studies S-180 sarcoma cells were used to induce a xenograft [19,20], and in the remaining study, PC3 prostate cancer cells were used [13]. The extracts (ethanol and chloroform) were both active against the sarcoma xenograft and inhibited growth, but the powder obtained from an *F. pinicola* ethanol extract showed no activity against the prostate cancer xenograft. The discrepancy in the results is thought to be due to the lack of bioavailability of the ethanol powder extract, as well as treatment at a later stage of disease [13].

In addition to the three studies described in Table 3, Choi et al. also carried out an animal experiment whereby rats received 0.83 g/kg of mushroom for two weeks following the administration of ethanol for two weeks [31]. Glutathione, glutathione peroxidase and catalase were all found to be significantly higher in the intervention versus the control group [31]. Glutathione is an anti-oxidant and, together with glutathione peroxidase and catalase, protects the cell against oxidative damage, and thus can exert an anti-tumour effect. 

In a rat model with diabetes induced by streptozotocin, *F. pinicola* treatment decreased glucose levels, restored insulin levels to nearly normal, and pancreatic tissue damage was ameliorated [33]. The alkali extract was more effective than the water extract at reducing the harmful effects of streptozotocin induced diabetes [33]. Enhanced glucose uptake, and therefore hyperglycaemia, is a metabolic characteristic of cancer cells, and therefore the link between diabetes and cancers [34] is not surprising. Although the in vivo study by Lee et al. focused on the impact of *F. pinicola* extracts on diabetes, the benefits could be extrapolated to the treatment of cancers. This hyperglycaemic effect, in the context of cancers, should be investigated further.

## 5. Limitations

The most obvious limitation of this review is the lack of certainty surrounding the identity of the specimens used in the studies we discuss. For example, Gao et al. 2017 state that *F. pinicola* is traditionally categorized as Reishi [19], yet Reishi is identified as *Ganoderma lucidum* (Lingzhi) [28]. It is therefore unclear as to whether Gao et al. studied *G. lucidum* or *F. pinicola* and the method of species identification is not stated. 

Another limitation includes the small number of in vivo studies performed. The fact that only three in vivo studies have been carried out, whereby the anti-cancer properties of *F. pinicola* were investigated, indicates the paucity of data and the need to carry out further studies in different cancer models.

## 6. Conclusions

In conclusion, further research is required to characterise the anti-cancer activities of *F. pinicola* as there is a paucity of data, particularly from in vivo and clinical studies. It would be useful to identify the bioactive components of *F. pinicola* and build on the research performed by Wang et al. and Gao et al. [19,26]. In particular, care must be taken to correctly identify each specimen using molecular techniques, prior to experimentation. Like many food components, *F. pinicola* has the potential to reduce the risk of disease. The advantage of investigating the anti-cancer benefits of *F. pinicola* is that the mushroom is not toxic as shown by anecdotal evidence over centuries, as well as in vivo studies. In addition, it is widely available and is affordable.

## Figures and Tables

**Figure 1 nutrients-12-00609-f001:**
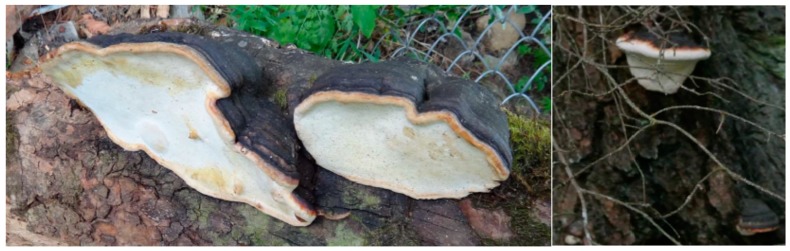
Basidomes of *Fomitopsis pinicola* in situ. These specimens were identified as *F. pinicola* by internal spacer region 2 sequencing [13]. (Permission was obtained from NZFocus to utilise this image.)

**Table 1 nutrients-12-00609-t001:** Extraction method and fraction or compound detected from *F. pinicola* specimens.

Citation	Extraction Method	Details of Method	Fraction/Compound/Concentration
Gao et al. 2017, [19]	Methanol	95% ethanol or methanol for 8 to 10 hours at room temperature, evaporated and washed in water, evaporated and lyophilized.	0.210 µM GAE/mg
	Hot water	Heated at 100 °C for 2 to 3 hours; 4 °C overnight; centrifuged and lyophilized.	0.185 µM GAE/mg
	Ethyl acetate	NT.	0.464 µM GAE/mg
	Petroleum ether	NT.	0.389 µM GAE/mg
Wu et al. 2014, [20]	Ethanol	Extracted three times with 50% ethanol or water for 24 h. The solutions were filtered, the solvent removed by distillation and the sample was lyophilised.	NT
Gao et al. 2017, [19]	Chloroform/ethanol	The specimens were dried and milled. Thereafter, the powder was homogenised in 95% ethanol at 45 °C and subjected to ultrasonic- assisted extraction. The extract was fractionated with chloroform, homogenised in ethanol, centrifuged and the supernatant was filtered.	Ergosterol (105 µg/mg) Pachymic acid (35.6 µg/mg) Dehydroeburicoic acid (2.5 µg/mg).
Kao, 2019; Kao et al. 2018; Kao et al. 2016, [13,21,22]	Whiskey/Rice wine (ethanol)	Submerged in whiskey or rice wine for six months, then freeze dried. The whiskey extract was fractionated 1:1:1:1 (water: methanol: ethanol: chloroform).	Aqueous fraction Organic fraction (56.4% and 43.6% total weight respectively)
Yoshikawa et al. 2005, [23]	Ethanol	Submerged in 70% ethanol for six weeks and separated into EtOAc and H_2_O portions. Thereafter, the EtOAc extract was fractionated using silica gel column chromatography, and some fractions were further fractionated using an HPLC.	Lanostane triterpenes: Fomitopinic acid A and B Lanostanoid glycosides: Fomitoside A-J
Keller et al. 1996, [24]	Dichloro-methane	Lanostenoid derivative. Seven triterpenes.	3α-(4-car boxymethyl-3-hydroxy-3-methylbutanoyloxy)-lanosta -8,24-dien-21-oic acid, polyporenic acid C, 3α-acetyloxylanosta-8,24-dien-21-oic acid, ergosta-7,22-dien-3/J-ol, 21-hydroxylanosta-8,24-dien- 3-one, pinicolic acid A, trametenolic acid B and pachymic acid21-oic acid

EtOAc—Ethyl acetate; GAE—gallic acid equivalents; HPLC—high-performance liquid chromatography; NT—not tested.

**Table 2 nutrients-12-00609-t002:** In vitro studies in which the anti-cancer properties of *F. pinicola* were investigated.

Citation	Cell Line *	Cancer Type	Type of Extract	Cell Viability 1000 µg/ml (%)	IC50^+^	Outcomes
Choi et al. 2007, [31]	N/A	N/A	Not specified	N/A	N/A	Increased anti-oxidant activity
	HeLa	Cervix	Water Ethanol	70.0 25.0	NT NT	
	HO-1	Melanoma	Water Ethanol	98.0 40–45	NT NT	
	SNU-354	Liver	Water Ethanol	65.0 35–45	NT NT	
	SNU-185	Liver	Water Ethanol	60.0 35–45	NT NT	
	SK-Hep3B	Liver	Water Ethanol	<82.0 <50.0	NT NT	
	Hep3B	Liver	Water Ethanol	<82.0 <40.0	NT NT	
	PLC/RF/5	Liver	Water Ethanol	95.0 <40.0	NT NT	
Wu et al. 2014, [20]	S-180 (mouse)	Sarcoma	Water Ethanol	78.9 17.2	NT NT	NT Increased CC3, APAF-1 and C-PARP
	HepG2	Hepatoma	Water Ethanol	96.6 28.7	NT NT	NT Increased CC3; NT; NT
	A549	Lung	Water Ethanol	97.0 7.1	NT NT	NT Increased CC3; NT; NT
	HCT-116	Colon	Water Ethanol	62.5 12.1	NT NT	NT Increased CC3; NT; NT
	MDA-MB-231	Breast	Water Ethanol	60.1 34.1	NT NT	NT Increased CC3; NT; NT
Gao et al. 2017, [19]	S-180 (mouse)	Sarcoma	FPKc	NT	36.2	Induced late stage apoptosis/decrease in MMP/DNA fragmentation
	HL-60	Leukemia	FPKc	NT	41.0	NT
	K562	Leukemia	FPKc	NT	98.9	NT
	U937	Leukemia	FPKc	NT	34.9	NT
	SMMC-7721	Hepatoma	FPKc	NT	246.2	NT
	Eca-109	Esophageal	FPKc	NT	169.7	NT
Wang et al. 2014, [26]	SW-480	Colon	FPKc	NT	190.3	Inhibits cell migration and induce apoptosis.
	SW-640	Colon	Ergosterol FPKc	NT NT	143.3	Induced cell apoptosis NT
Kao et al. 2018; Kao et al. 2016 [21,22]	PC3	Prostate	WhE	NC	NT	Upregulation of pro-apoptotic genes, and down-regulation of anti-apoptotic genes. Significant changes in gene expression associated with cell-cycle pathways, amongst others.
	DU145	Prostate	WhE	NC	NT	Significant changes in gene expression associated with cell-cycle pathways, amongst others.
Yoshikawa et al. 2005, [23]	N/A	N/A	Ethanol (fomitopinic acid and fomitosides)			Anti-inflammatory activity in response to COX 1 and 2.

* All cell lines are of human origin, unless otherwise stated. ^+^ IC50 was measured at 72 h in µg/ml. Abbreviations: APAF1-apoptotic peptidase activating factor 1; CC3—cleaved caspase 3; COX-cyclooxygenase; C-PARP—cleaved-poly ADP ribose polymerase; FPKc—*F. pinicola* chloroform extract; IC50—half maximal inhibitory concentration; MMP—mitochondrial membrane potential; N/A-not applicable; NC—not comparable (reported in µl); NT—not tested/not reported; WhE—whiskey extract.

**Table 3 nutrients-12-00609-t003:** In vivo studies in which the anti-cancer properties of *F. pinicola* were investigated.

Citation	Study Design	Treatment	Type of Extract	Delivery of Extract	Outcomes
Wu et al. 2014, [20]	Balb/c male mice with S180 xenograft	1.5–5 g/kg; 3 and 7 days prior to xenograft	Ethanol extract	Dietary supplement	Inhibition of tumour growth (growth inhibitory ratio = 54% compared to control) and prolonged survival (40% survival in the control group, and 60%–70% survival in the intervention groups at day 30).
Gao et al. 2017, [19]	ICR mice with S180 xenograft	200 mg/kg; 7 days prior to xenograft	FPKc	Inoculated subcutaneously	Inhibition of tumour growth (inhibition rate = 47.7% compared to control) and prolonged survival (control group–survival ranged from 12 to 15 days, and the intervention group, survival ranged from 15 to 19 days.
Kao, 2019, [13]	Rag-1 male mice with PC3 xenograft	1 g/kg once the tumour had reached 200 mm^3^	Ethanol-based extract as a powder	Oral gavage	Dose was tolerated. No noticeable effect.

Abbreviations: FPKc—chloroform extract of *F. pinicola;* ICR—institute of cancer research.

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
