# Peer review of "Characterisation of Extracts and Anti-Cancer Activities of Fomitopsis pinicola"

_nutrients, 2020, doi:10.3390/nu12030609_

Round 1

Reviewer 1 Report

The authors tried to review about 'characterisation of extracts and anti-cancer activities of Fomitopsis pincola'. So they showed the its fraction and compound by extraction methods. Also showed their effect of in vitro an in vivo. I wondered major component and effective component of Fomitopsis pincola. Also, I wondered which component showed anti-cancer activities.

I suggest that you describe the detailed ingredients about Fomitopsis pincola, then article is more valuable.

Author Response

The authors tried to review about 'characterisation of extracts and anti-cancer activities of Fomitopsis pincola'. So they showed the its fraction and compound by extraction methods. Also showed their effect of in vitro an in vivo. I wondered major component and effective component of Fomitopsis pincola. Also, I wondered which component showed anti-cancer activities.

I suggest that you describe the detailed ingredients about Fomitopsis pincola, then article is more valuable.

Thank you for taking the time to review my manuscript. I have included additional text to address your comments. Information regarding active components is largely missing from the literature. Where this information is available, it has been included in table 2 and mentioned in the text (lines 130-140). The legend for table 1 has been modified so as not to be misleading.

Reviewer 2 Report

This is an interesting and generally well written article.  I provide the following suggestions as ways to improve the manuscript.

Line 43-47:  On 10-Feb-20 I did a search for Fomitopsis pinicola on the Wed of Knowledge. Three were 232 references (although only 12 that also mentioned cancer). This review only cites 33 articles. Clearly not all of the 232 are relevant. However, can you provide some additional details about what your search details were, how many articles your search found, how many were culled for what reason and how many were included in this review etc? Line 92:  Can you add examples of what medicinal uses F. pinicola has been used for? Line 92/93:  How were the mushrooms prepared and used for traditional medicinal use?  Maybe a table could be added to provide information pertaining to my Q 2 and Q3. Table 2:  The information in the 5th column of the table is a little bit confusing. I suggest you consider splitting this into two columns. Therefore, you would have one as cell viability the other as IC50; there will obviously be a lot of NAs but to have different data in the same column is a touch confusing in my opinion. Table 3:  Can you provide more specific detail in the Outcomes column?  For example, what level of inhibition and prolonged survival? Conclusions:  Can the author draw any conclusions about the potency of the anti-cancer properties of F. pinicola? Can comparisons be made to other species? The author indicates that further research is required, which is true, but providing some opinion of the value of doing that research versus exploring the anti-cancer potential of other material would be useful.

Author Response

This is an interesting and generally well written article.  I provide the following suggestions as ways to improve the manuscript.

Thank you for taking the time to review my manuscript and for your constructive comments.

Line 43-47:  On 10-Feb-20 I did a search for Fomitopsis pinicola on the Wed of Knowledge. Three were 232 references (although only 12 that also mentioned cancer). This review only cites 33 articles. Clearly not all of the 232 are relevant. However, can you provide some additional details about what your search details were, how many articles your search found, how many were culled for what reason and how many were included in this review etc?

Additional information has been included. Please see lines 51-52 and 55-60.

Line 92:  Can you add examples of what medicinal uses F. pinicola has been used for?

This information has been added to line 112 (reference 16).

Line 92/93:  How were the mushrooms prepared and used for traditional medicinal use?  Maybe a table could be added to provide information pertaining to my Q 2 and Q3.

Almost no accessible information is available regarding the traditional uses of F. pinicola. https://reishiandrosesbotanicals.com/2015/04/ discuss preparation of F. pinicola, referencing Robert Rogers (2011 and 2014), as well as Grienke et al 2014. However, these publications refer to the use of specimens sourced from North America, and therefore we know that it wasn’t actually F. pinicola that was used. I have included text (Line 108-110) regarding a general reference to preparation.

Table 2:  The information in the 5th column of the table is a little bit confusing. I suggest you consider splitting this into two columns. Therefore, you would have one as cell viability the other as IC50; there will obviously be a lot of NAs but to have different data in the same column is a touch confusing in my opinion.

Column 5 has been divided into two columns as suggested, thus removing potential for confusion.

Table 3:  Can you provide more specific detail in the Outcomes column?  For example, what level of inhibition and prolonged survival?

Additional information has been added to the “outcomes” column as suggested.

Conclusions:  Can the author draw any conclusions about the potency of the anti-cancer properties of F. pinicola? Can comparisons be made to other species? The author indicates that further research is required, which is true, but providing some opinion of the value of doing that research versus exploring the anti-cancer potential of other material would be useful.

Thank you for your comment. I have strengthened the conclusion so that it is less vague and more meaningful.

Reviewer 3 Report

In this review, the author provided an up-to-date summary of the main findings of the research work concerning the characterization of Fomitopsis pinicola and its anticancer activities. This review contains a good amount of information, and it deals with an area of research that is beginning to come in for special interest from researchers and clinicians. I only have several minor comments:

Line 89-93 on Page 3: This paragraph is irrelevant to the taxonomy of F. pinicola, and should be put under “Introduction”.

In Line 98 on Page 3, the author stated that “”very little research has been carried out on F. pinicola”. However, in Line 114-115 on Page 4, it was stated that “Numerous in vitro studies on 114 cancer cell lines have been performed, ...”. It sounds contradictory. Also, there are only 7 in vitro studies in Table 2. I don’t think that is considered “numerous”.

Author Response

In this review, the author provided an up-to-date summary of the main findings of the research work concerning the characterization of Fomitopsis pinicola and its anticancer activities. This review contains a good amount of information, and it deals with an area of research that is beginning to come in for special interest from researchers and clinicians.

Thank you for taking the time to review my manuscript.

I only have several minor comments:

Line 89-93 on Page 3: This paragraph is irrelevant to the taxonomy of F. pinicola, and should be put under “Introduction”.

This paragraph has been moved to the Introduction as suggested (Lines 44-49).

In Line 98 on Page 3, the author stated that “”very little research has been carried out on F. pinicola”. However, in Line 114-115 on Page 4, it was stated that “Numerous in vitro studies on 114 cancer cell lines have been performed, ...”. It sounds contradictory. Also, there are only 7 in vitro studies in Table 2. I don’t think that is considered “numerous”.

Thank you for bringing this point to my attention. The word “numerous” was inappropriate and the sentence has been changed. (Lines 151-153)